# The Relationship between Giftedness and Sex and Children’s Theory of Mind Skills and Social Behavior

**DOI:** 10.3390/children11020253

**Published:** 2024-02-16

**Authors:** Abdullah Bozkurt, Zekai Ayık

**Affiliations:** 1Department of Child and Adolescent Psychiatry, Ataturk University, Erzurum 25240, Turkey; 2Department of Special Education, Harran University, Şanlıurfa 63290, Turkey; zekaiayik@harran.edu.tr

**Keywords:** gifted, theory of mind, social behavior, sex

## Abstract

Background: Theory of mind (ToM), the ability to recognize the mental states and emotions of others, is central to effective social relationships. Measuring higher-order ToM skills in gifted children may be a useful way to identify the tendency to experience difficulties in social behavior. The aim of this study was to investigate the relationship between intelligence and sex in children using ToM and social behavior measures. Methods: Children aged 10–12 years constituted both the gifted (*n* = 45) and non-gifted (*n* = 45) groups. The participants were assessed for prosocial behaviors and peer problems using the subscales of the Strength and Difficulties Questionnaire and in terms of ToM using the Reading the Mind in the Eyes Test-Child Version (RMET-C) and the Faux Pas Recognition Test-Child Version (FPRT-C). Results: ToM test results were higher in gifted children and girls. Peer problems were lower in gifted children. Prosocial behavior was higher in girls. No relationship was determined between ToM tests and peer problems or prosocial behavior in gifted children, but such a relationship was observed in the non-gifted group. Conclusions: This study shows that gifted children with high cognitive skills also possess superior social cognition skills. Advanced ToM skills in gifted children may be important to supporting their social and cognitive development. The differences between boys and girls should be considered in educational interventions applied to children in the social sphere.

## 1. Introduction

Gifted individuals demonstrate advanced intellectual development in comparison to their social, emotional, and physical development trajectories that are typically similar to their chronological age peers [1]. As these development fields are interconnected, superiority in intellectual development may be expected to affect other developmental trajectories and skills among gifted individuals [2]. Social skills are one of those competencies thought to be affected by high intellectual abilities. Cross et al. suggested that social skills bestow abilities to assess experiences in the social context, to evaluate actions in terms of the degree to which they lead to the desired end results, and ultimately to adapt behavior accordingly [3]. Furthermore, several studies [2,4] have suggested that social development is a crucial phenomenon for the healthy functioning of gifted individuals and the development of their gifts and talents, and that these are affected by intellectual abilities. In light of this importance, social competencies are viewed as predictors of success among gifted students [4]. Since the talent development process is affected by the social and interpersonal skills of gifted students, the social skills of such individuals have attracted the attention of researchers in the field of giftedness for several decades.

Prosocial behavior, a social skill which includes actions intended to benefit others, has become a topic of interest in various fields. Research has shown that prosocial behavior is influenced by various factors, such as empathy, genetic predispositions, and social contexts [5,6,7]. Findings also highlight the positive effects of prosocial behavior on children’s academic achievement, cognitive development, and social relationships [8,9,10]. Gifted children have been shown to exhibit high levels of empathy and sensitivity, thus contributing to their prosocial behaviors [11]. Promoting empathy and prosocial behaviors in gifted children is regarded as essential for their social and emotional development [8]. Intervention programs aimed at promoting prosocial behaviors in children were found to show significant effectiveness for these behaviors [12,13]. A meta-analysis study reported that programs targeting prosocial behavior and empathy-related skills were more effective on social and academic outcomes when implemented at earlier ages and included more empathy skills [14]. The empathy training program was found to have lasting effects on gifted adolescents with low empathy scores [15]. In a recent systematic review, experiential learning programs among learning programs were determined to be more effective in developing empathy [16]. A systematic review by Cheang et al. suggested mindfulness-based interventions increase empathy in children and adolescents [17]. Implementing these programs for psychosocial competence in gifted children may be helpful.

In order to ensure appropriate social interaction and communication, one needs to recognize that others have beliefs that differ from one’s own and to be able to behave in relation to these beliefs. This ability, known as theory of mind (ToM), is described as recognizing other people’s mental states and emotions and can be regarded as of pivotal importance to successful social interactions [18]. Underlying mental states including beliefs and intentions are known to be significantly associated with children’s behaviors that function successfully in their relationships with peers. Children who are unable to grasp that actions result from underlying mental states may be less successful in their peer relationships. Such children may be exposed to continuing peer problems and, thus, to long-term social adjustment difficulties [19]. The relationships between ToM abilities and children’s social interactions therefore require investigation. Since poor social competence in middle childhood is a significant indicator of subsequent mental health problems and less successful academic achievement, it is vitally important to understand variations in the ability to build, manage, and maintain social relationships in this period [20,21].

The assumption that diversity in ToM is important for children’s social lives has increased interest in their understanding of the minds of others [22]. Devine and Hughes have shown that individual differences in ToM are related to loneliness and dissatisfaction in relationships with classmates [23]. A meta-analysis suggested that children with better ToM were more likely to be accepted and less likely to be rejected by their peers [24]. Banerjee et al.’s study provides evidence not only of a contemporaneous relationship between ToM and children’s social experiences with peers at school, but also of a longitudinal relationship [25].

Children with advanced ToM abilities tend to engage in more complex social and communicative behaviors [26]. High ToM performance is predicted to benefit social competence over and above any effect of general cognitive factors [27]. In their study of school-age children, Liddle et al. observed a correlation between the degree of social competence and higher-level ToM performance. This suggests that more nuanced, recursive mental state understandings are crucial to social behavior [28]. Measuring higher-order ToM skills in gifted children may be a valuable means of identifying a tendency to experience difficulties in social behavior.

Studies of the social abilities of gifted students have reported inconsistent results. For example, some studies [29,30] have found that gifted students possess better social abilities than their peers of average intelligence. This is because their advanced cognitive abilities enable them to accurately assess social settings, and to select and implement courses of action which result in the desired consequences [31]. However, other studies [31,32] show that gifted children experience difficulties in social adaptation and interpersonal relationships, resulting in low-level social skills. Eren et al. compared gifted children and adolescents across a wide age range (9–18 years) with a non-gifted group and found no difference in the parents’ prosocial behavior and peer relationship scores [33]. The fact that in terms of methodology the researchers methodologically used self-generated questions, valid and reliable scales, and parent or self-report scales in those studies may account for the differences in the findings.

The relationship between gifted children’s ToM skills and social situations is a highly important one. To the best of our knowledge, no previous studies have compared the high-level ToM skills of gifted children with those of their peers with typical development. The relationships between social relationships and ToM in gifted children have not been clearly demonstrated, and may differ between the sexes. These relationships are essential when designing educational programs for gifted children and may be related to different needs in boys and girls.

The main purpose of this study was to investigate the relationship between intelligence and sex in children using ToM and social behavior measures. The specific aim was to determine differences in ToM and social behavior by comparing gifted and non-gifted children. We also set out to evaluate whether there are sex differences in ToM and social behavior between gifted and non-gifted children and to determine the relationships between ToM and social behavior in both groups. We hypothesized that gifted children would exhibit better ToM skills and social behavior than non-gifted children and that girls would exhibit better ToM and social behavior than boys.

## 2. Materials and Methods

### 2.1. Participants

The study was conducted with 45 gifted (22 boys, 23 girls) and 45 non-gifted (18 boys, 27 girls) children aged 10–12 years. The gifted population consisted of students attending Science and Arts Center (SAC) pull-out enrichment schools for gifted individuals. Students at SACs also attend mainstream schools. In order to select participants from similar socioeconomic backgrounds, non-gifted students were selected from the five general schools attended by the selected gifted students. SACs are public institutions found in cities all over Turkey intended to provide supplementary education for gifted students. Students are selected in three stages. In the first stage, they are nominated by their classroom teacher. In the second stage, students take a valid, reliable, and standardized intelligence test applied during individual examinations. The current intelligence test is the Anadolu-Sak Intelligence Scale (ASIS) developed by [34]. Students who pass the screening exam and those who score 130 or above on the ASIS test for each grade level and for each talent area are enrolled in a SAC. In the third stage, students take a screening exam which identifies the areas of talent, and they are educated at SACs according to these talent fields. The non-gifted population in this study consisted of students who were not nominated by their classroom teachers for tests of SACs or nominated by scoring below 130 on the ASIS test.

Participants with no previous chronic medical illness or psychiatric diagnosis and who voluntarily agreed to participate in the study were assessed for prosocial behaviors and peer problems using the subscales of the Strength and Difficulties Questionnaire (SDQ) and for ToM using the Reading the Mind in the Eyes Test-Child Version (RMET-C) and the Faux Pas Recognition Test-Child Version (FPRT-C).

### 2.2. Measures

#### 2.2.1. Strength and Difficulties Questionnaire

The SDQ is a 25-item scale used for assessing mental health problems in children aged 4–16 years [35]. As a mental health screening tool, the SDQ has been widely used in research and clinical settings across numerous different countries and cultures [36]. The peer problems and prosocial behavior subscales were used in this study. The Cronbach Alpha values in this study were 0.72 and 0.80, respectively. The scale was adapted into Turkish by Güvenir et al. [37].

#### 2.2.2. ToM Tests

Reading the Mind in the Eyes Test-Child Version (RMET-C). This advanced ToM test assesses the individual’s ability to make inferences about another’s mental state simply by looking at eye photographs. The participants were shown various pictures of the eye area accompanied by four labels referring to different feelings. They were then asked to select which word best described the eyes in the photograph [38]. The Turkish-language version was used in the present study [39]. The Cronbach Alpha value in this study was 0.73.

Faux Pas Recognition Test-Child Version (FPRT-C). Baron-Cohen et al. created and applied the Faux Pas Test to assess higher mental attributions [40]. Recognizing a faux pas is commonly regarded as the most difficult developmental skill and as a sensitive evaluation instrument for ToM. A faux pas happens when an individual says something he or she should not have said without realizing or being aware of it. After listening to a narrative, the child answered four comprehension questions. In order to identify a faux pas, the child must correctly respond to all inquiries, answer a comprehension question, and understand that the faux pas resulted from an erroneous belief. In the control stories, the child must establish that no faux pas occurred. Any of these questions being answered incorrectly will result in a score of zero for that particular narrative. The lowest possible score on the entire test is 0 and the highest possible score 20, with a range of 0–10 for the faux pas stories and 0–10 points for the control stories. The Turkish-language version was used in this study [41]. The Cronbach Alpha value in this study was 0.77.

#### 2.2.3. Anadolu Sak Intelligence Scale

The Anadolu-Sak Intelligence Scale (ASIS) is an individually implemented intelligence test battery developed for Turkish-speaking children aged 4 to 12 and normed in Turkey in 2016 [34,42]. ASIS is theoretically based on the Cattell–Horn–Carroll model of intelligence and cognitive abilities [43]. It consists of 256 items, seven subtests, and three factors. Several researchers have investigated the reliability and validity of ASIS [34,42,44]. The validity of ASIS was studied and confirmed using exploratory and confirmatory factor analyses. Sak et al. determined correlations between ASIS scores and the UNIT and the RIAS intelligence test scores ranging from 0.50 to 0.82 [34]. Studies investigating the reliability of ASIS demonstrate that the internal consistency of the subtests and the component scores range from 0.81 to 0.94 in the standardization sample [45]. Inter-coder reliability has been reported at 0.96 and above [34]. Test–retest reliability for the subtests ranges from 0.66 to 0.85, and between 0.91 and 0.95 for the factors [44].

### 2.3. Procedure and Data Collection

The study data were collected in SACs and general public schools using the same procedure for both the gifted and non-gifted groups. The data were collected between 9 September and 10 October 2023. The effect size (Cohen’s d) was calculated as 0.64, with 80% power and a 95% confidence interval, analysis showing that 40 children in each group should be included in the study [46]. Based on a 10% probability of dropouts in each group, a minimum of 88 participants with 44 patients in each group was required. The teacher was first informed about the process and the content of the data collection tools. Written informed consent forms were obtained from the children and parents. A socioeconomic demographic form was completed for students consenting to take part, and the RMET-C and FPRT-C were administered by an educator trained in ToM testing. The children also completed the SDQ scale. Approval for the study was granted by the Harran University Social Science Ethical Committee (number E-76244175-050.01.04-230032).

### 2.4. Statistical Analysis

The collected data were analyzed on SPSS version 26.0 software (SPSS Inc., Chicago, IL, USA). There were no missing data for any of the variables. Descriptive statistics were applied for the sociodemographic data. Parametric methods were applied for normally distributed variables. A chi-squared test was performed to assess differences between the study and control groups in terms of sex distribution. Student’s t or the Mann–Whitney U test were applied in comparing differences between two groups in line with their distribution. The Pearson test was implemented to calculate correlation coefficients and significance between two normally distributed parameters. The Spearman test was used to investigate the correlation coefficients and significance of non-normally distributed parameters. In addition to the whole sample, the same analyses were performed separately for both sexes to assess the potential sex-specific association between the targeted measures and groups. Two-way analysis of variance (two-way ANOVA) was applied to determine the main effects of group (gifted and non-gifted) and sex (male and female) and the interaction effect of group × sex on the scale and ToM test scores. The effect size was calculated according to Cohen’s d and Cramer’s V statistics. Statistical significance was set at *p* < 0.05.

## 3. Results

No significant sex or age differences were observed between the gifted and non-gifted groups. Peer problem scores were statistically significantly higher in the non-gifted group than in the gifted group at medium effect size, but no significant difference was found in prosocial behavior scores. Boys in the gifted group had significantly fewer peer problems (large effect size) and more prosocial behaviors (medium effect size) than boys in the non-gifted group, while no significant difference was found between the girls in the two groups. ToM tests were statistically significantly higher for boys in the gifted group than in the non-gifted group at large effect size; only the FPRT-C scores were higher for girls in the gifted group. ToM test results were statistically significantly higher in the gifted group than in the non-gifted group at medium and large effect sizes (Table 1).

Girls from the non-gifted group exhibited significantly lower peer problem scores and higher prosocial behavior, RMET-C, and FPRT-C scores than boys, although no significant differences were determined between boys and girls from the gifted group (Table 2).

The group effect (F_1,86_ = 8.237, *p* = 0.005) had a medium significant effect on peer problems, while sex (F_1,86_ = 3.301, *p* = 0.073) was not significant. The interaction of group and sex (F_1,86_ = 3.991, *p* = 0.049) also exerted a small to medium significant effect on peer problems. The gifted group registered lower peer problem scores. In the non-gifted group, peer problems were higher in the boys. Sex (F_1,86_ = 4.300, *p* = 0.041) had a small to medium significant effect on prosocial behavior scores, while the group effect (F_1,86_ = 0.613, *p* = 0.436) was not significant. Group and sex interaction (F_1,86_ = 5.837, *p* = 0.018) resulted in medium significant differences in prosocial behavior scores. Girls registered higher prosocial behavior scores. In the non-gifted group, the prosocial behavior score was lower among the boys. The group effect (F_1,86_ = 8.374, *p* = 0.005) and sex (F_1,86_ = 5.148, *p* = 0.026) had a medium significant effect on RMET-C scores. However, the interaction of group and sex (F_1,86_ = 1.563, *p* = 0.215) was not significant on RMET-C scores. The group effect (F_1,86_ = 26.832, *p* < 0.001) and sex (F_1,86_ = 10.127, *p* = 0.002) exerted large and medium significant effects on FPRT-C scores, respectively, although the interaction of group and sex (F_1,86_ = 1.923, *p* = 0.169) was not significant on FPRT-C scores. ToM test scores were higher in the gifted group and in girls (Table 3).

In the non-gifted group, peer problems and prosocial behavior scores were correlated with RMET-C and FPRT-C scores, but there was no correlation in the gifted group (Table 4).

## 4. Discussion

ToM test results in this study were higher among the gifted children and girls. Peer problems were lower among the gifted children but higher in boys from the non-gifted group, while no sex difference was observed among the gifted children. Prosocial behavior was higher in girls. While no sex differences were observed in the gifted children, prosocial behavior was lower in boys from the non-gifted group. No relationship was determined between ToM tests and peer problems or prosocial behavior in the gifted children, but such an association was observed in the non-gifted group.

The higher medium and large effect sizes of the ToM test results in gifted children indicate that this difference is practically significant. ToM skills are closely linked to children’s social relationships [26]. Children with advanced ToM skills have a greater likelihood of recognizing and appreciating their peers’ thoughts, feelings, and perspectives, making them more socially competent and popular in their peer groups [47]. Children with high ToM skills are more likely to engage in learning activities such as explaining and demonstrating new concepts to their peers [2]. This suggests that ToM is involved in children’s ability to understand and communicate information effectively [48]. The high level of ToM skills in gifted children determined in the present study may contribute to these strategies. It may also contribute to their success in social situations by generating valuable strategies and overcoming difficulties. To the best of our knowledge, no previous study has evaluated high-level ToM in gifted children. The current study thus fills this important gap in the literature.

In the literature, it is reported that girls have better ToM skills than boys [23]. This study’s finding that ToM test results were higher in the large effect size for girls reinforces the previous literature. The absence of group differences on the RMET-C test among the girls in this study may be explained by the high performance of girls with average intelligence on the RMET-C and the ceiling effect of the test. While affective ToM is assessed in the RMET-C test, both affective and cognitive ToM are assessed in the FPRT-C test [49]. The assessment of both affective and cognitive domains in the FPRT-C test is a more complex and advanced ToM skill than other ToM skills [50]. The superiority of gifted girls in ToM tasks is revealed through more challenging tests such as the FPRT-C test.

Whether high intelligence constitutes a risk or a protective factor for social behaviors is still controversial. While there are reports that gifted children exhibit better socio-emotional adaptation, more cooperative behaviors towards their peers, and fewer behavioral problems compared to their peers, other studies have reported social-emotional problems or no difference [34,51,52,53]. The gifted group in the present study having fewer peer problems and exhibiting more prosocial behavior in early adolescence suggests that being gifted is a protective factor in social behaviors. Higher ToM skills may also contribute to this process. The present study’s finding of higher prosocial behaviors in gifted boys with small to medium effect sizes should be confirmed in a larger sample or with further studies. Programs aimed at developing empathy skills in gifted children have been shown to yield positive results [15]. Since prosocial behaviors play a crucial role in social adjustment to and academic success in school [54], the education of gifted students should be reinforced with high prosocial behaviors exhibited by gifted individuals.

Studies have shown sex differences in peer problems and prosocial behavior and that girls generally exhibit higher levels of prosocial behavior and have better relationships with their peers than boys [55,56]. In this study, and consistent with the literature, prosocial behavior was higher in girls than boys. While no sex difference was determined among the gifted children, the finding of lower prosocial behavior in boys from the non-gifted group suggests that giftedness improves social responsibility and sensitivity among boys. The differences between boys and girls may need to be considered in terms of the effectiveness of educational interventions applied to children in the social sphere.

ToM skills, which are thought to represent a protective factor in social relationships, were associated with peer relationships and prosocial behavior in the non-gifted group but not in the gifted group. It has been suggested that children with high IQs may not exhibit socio-emotional problems until later in childhood [57]. Considering that problems in social behaviors may emerge at later ages, there may be a relationship between ToM skills and social behaviors in gifted children at older ages. Cognitive skills other than ToM, such as executive functions and pragmatic skills, are related to peer relationships in children [58,59]. In addition, Holmes et al. reported that the effect of executive functions on peer relations decreases as children progress into adolescence [60]. Social behaviors in gifted children may be related to other cognitive abilities, such as executive functions and language skills. The relationship between ToM and peer relationships may decrease during adolescence, as shown in executive functions.

This study is important since there has been no previous evaluation of high-level ToM skills in gifted children in the literature. The strong points of this study are that the age range was narrow, sex differences were evaluated, the groups were composed of individuals with similar sociocultural characteristics, and the study involved a heterogeneous sample representing gifted children in various areas of ability, including science, mathematics, technology, and arts.

However, there are also a number of limitations to this study that need to be considered. First, its cross-sectional study design prevented us from assessing the direction of the reported association between ToM, sex, and social behaviors. The sample size was limited, and further research is now needed if our results are to be confirmed and generalized. Another limitation is that we focused on a single self-report measure when assessing social behaviors. Future studies should adopt a multi-informant approach to obtain a robust index of social ability. This study evaluated peer relations, prosocial behaviors, and ToM skills. Also, taking social relations such as parental relations, relations with authority, and sibling relations into account may reveal the situation in social fields more clearly. Two different areas of cognition have been defined as hot and cold cognition from the perspective of understanding cognitive processes [61]. In the present study, ToM tests mainly assessed hot cognition, and cold cognition areas, such as executive functions (e.g., working memory), were not examined. The literature emphasizes the relationship between social behavior and other cognitive abilities, such as executive functions [58]. At the same time, Di Tella et al. determined no relationship between emotional/cognitive ToM and executive functions, while Stone and Gerrans espoused the opposite view [62]. Further studies evaluating ToM skills and executive functions in gifted children may help clarify this issue. Researchers have proposed that children with high IQs may not exhibit socio-emotional problems until later in childhood [57]. Longitudinal studies are now needed to understand how this relates to problems that increase later in adolescence. The concept of giftedness to a large extent relies on an IQ-based categorization. Children with the same IQ levels can still possess very different intellectual profiles. Children with higher verbal intelligence than non-verbal intelligence may exhibit a completely different pattern of social behavior and ToM skills than those with higher non-verbal intelligence. It may be more appropriate to plan future research into social behavior patterns and ToM skills by also considering interpersonal variability. Different behavioral patterns have been reported among individuals with “high” and “low” gifted profiles [2]. Future research may need to evaluate social behaviors and ToM in “high” and “low” giftedness levels. Gifted children with a history of psychiatric illness were excluded from the present study. Autism spectrum disorder, learning disabilities, and attention deficit hyperactivity disorder may affect social behavior and ToM in gifted children. Future research should consider such additional problems.

This study is important in showing that gifted children with high ToM skills also exhibit higher abilities in social cognition. Advanced ToM skills in gifted children can be considered an important cognitive ability for the promotion of social and cognitive development. Sex-specific differences observed between boys and girls should be recognized and integrated into educational interventions for children in the social sphere.

## Figures and Tables

**Table 1 children-11-00253-t001:** Age, sex, ToM tests, and SDQ subscale scores in the gifted and non-gifted children.

	Boys		t or z or χ²	*p*	*d*	Girls		t or z or χ²	*p*	*d*	Total Sample		t or z or χ²	*p*	*d/V*
	Gifted children (*n* = 22)	Non-Gifted (*n* = 18)				Gifted children (*n* = 23)	Non-Gifted (*n* = 27)				Gifted children (*n* = 45)	Non-Gifted (*n* = 45)			
Sex (boy/girl)	22/0	18/0	-	-		0/23	0/27	-	-		22/23	18/27	0.720	0.396	0.089
Age (years)	11.7 ± 0.5	11.5 ± 0.5	1.337	0.189	0.425	11.6 ± 0.6	11.6 ± 0.6	−0.115	0.909	0.033	11.6 ± 0.6	11.5 ± 0.5	0.711	0.481	0.150
SDQ															
Peer problems	2.3 ± 1.5	4.2 ± 1.9	−3.437	0.001	1.092	2.4 ± 2.1	2.7 ± 1.7	−0.810	0.418	0.229	2.4 ± 1.8	3.3 ± 1.9	−0.244	0.017	0.515
Prosocial behavior	8.1 ± 1.6	6.8 ± 2.1	2.127	0.040	0.676	8.0 ± 2.1	8.7 ± 1.6	−0.994	0.320	0.281	8.1 ± 1.9	7.9 ± 2.0	−0.190	0.849	0.040
ToM tests															
RMET-C	20.0 ± 2.3	17.2 ± 3.2	3.178	0.003	1.009	20.7 ± 2.9	19.6 ± 3.8	1.133	0.263	0.322	20.4 ± 2.6	18.7 ± 3.7	2.515	0.014	0.530
FPRT-C	15.9 ± 2.2	12.5 ± 2.6	4.403	<0.001	1.399	16.9 ± 1.9	14.8 ± 2.7	2.927	0.006	0.808	16.3 ± 2.1	13.9 ± 2.8	4.545	<0.001	0.958

SDQ—Strength and Difficulties Questionnaire; ToM—theory of mind; RMET-C—Reading the Mind in the Eyes Test-Child Version; FPRT-C—Faux Pas Recognition Test-Child Version; *d*/V—effect size.

**Table 2 children-11-00253-t002:** A comparison of SDQ subscales and ToM tests between gifted and non-gifted boys and girls.

	Gifted					Non-Gifted				
	Boys (*n* = 22)	Girls (*n* = 23)	t or z	*p*	*d*	Boys (*n* = 18)	Girls (*n* = 27)	t or z	*p*	*d*
SDQ										
Peer problems	2.3 ± 1.5	2.4 ± 2.1	−0.150	0.881	0.045	4.2 ± 1.9	2.7 ± 1.7	−2.427	0.015	0.724
Prosocial behaviors	8.1 ± 1.6	8.0 ± 2.1	0.140	0.889	0.042	6.8 ± 2.1	8.7 ± 1.6	2.849	0.004	0.809
ToM tests										
RMET-C	20.0 ± 2.3	20.7 ± 2.9	1.088	0.277	0.323	17.2 ± 3.2	19.6 ± 3.8	2.187	0.029	0.652
FPRT-C	15.9 ± 2.2	16.9 ± 1.9	1.603	0.109	0.478	12.5 ± 2.6	14.8 ± 2.7	2.701	0.007	0.805

SDQ—Strength and Difficulties Questionnaire; ToM—theory of mind; RMET-C—Reading the Mind in the Eyes Test-Child Version; FPRT-C—Faux Pas Recognition Test-Child Version; *d*—effect size.

**Table 3 children-11-00253-t003:** Two-way analysis of variance (ANOVA) applied to the main effects of group and sex, and the group × sex interaction effect on scale scores and ToM test results.

	MS	F	η^2^_p_	MS	F	η^2^_p_	MS	F	η^2^_p_
	Group			Sex			Group × Sex Interaction		
SDQ									
Peer problems	28.066	8.237 **	0.087	11.247	3.301	0.037	13.599	3.991 *	0.044
Prosocial behaviors	2.205	0.613	0.007	15.483	4.300 *	0.048	21.015	5.837 *	0.064
ToM tests									
RMET-C	85.044	8.374 **	0.089	52.279	5.148 *	0.056	15.868	1.563	0.018
FPRT-C	154.204	26.832 **	0.238	58.199	10.127 **	0.105	11.051	1.923	0.022

SDQ—Strength and Difficulties Questionnaire; ToM—theory of mind; RMET-C—Reading the Mind in the Eyes Test-Child Version; FPRT-C—Faux Pas Recognition Test-Child Version; * *p* < 0.05. ** *p* < 0.01.

**Table 4 children-11-00253-t004:** Correlations between peer problems and prosocial behavior scores and ToM test scores in gifted and non-gifted children.

	ToM Tests		SDQ	
			Peer Problems	Prosocial Behaviors
Gifted	RMET-C	r *p*	−0.032 0.833	−0.116 0.450
	FPRT-C	r *p*	−0.278 0.064	0.227 0.134
Non-Gifted	RMET-C	r *p*	−0.445 0.002	0.464 0.001
	FPRT-C	r *p*	−0.471 0.001	0.346 0.020

SDQ—Strength and Difficulties Questionnaire; ToM—theory of mind; RMET-C—Reading the Mind in the Eyes Test-Child Version; FPRT-C—Faux Pas Recognition Test-Child Version.

## Data Availability

The data presented in this study are available on request from the corresponding author.

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
