# Peer review of "The Relationship between Giftedness and Sex and Children’s Theory of Mind Skills and Social Behavior"

_children, 2024, doi:10.3390/children11020253_

Round 1

Reviewer 1 Report

Comments and Suggestions for Authors

Review of “The impacts of giftedness and sex on children’s Theory of Mind skills and social behavior”

The authors of the present study aim at providing evidence for giftedness and sex influences on childrens’ Theory of Mind skills and social behavior in a sample of 45 gifted and 45 non-gifted Turkish students. I provide several minor and major points below (the comments are not ordered according to relevance), that should be addressed, before this ms. can be reconsidered for publication:

-        The authors describe, that stage three of the giftedness assessment is based on “a valid, reliable, and standardized intelligence test”, namely the Anadolu-Sak intelligence scale. Most readers of this paper will never have heard about this scale (at the very least, I haven’t) and providing a non-dated reference (line 99) to a test manual/book chapter (???) is not informative either. This is a minor point, because in case of test unreliability or non-validity of the used test, any real existing effect would become harder to detect due to the increase of the signal-to-noise ratio. However, the authors should fill the reader in at least about the defining characteristics of this test and provide a proper citation (please also provide information about the used test norms including the standardization year).

-        Please provide Cronbach alpha from your test administrations of both SDQ and TOM questionnaires/tests.

-        Move participant descriptives from the results to the participants section.

-        You should provide the readers with the information in which way groups differed from each other when describing results. It is not enough to indicate that there were differences if the effect direction (e.g., larger values of gifted vs. non-gifted individuals) is not provided.

-        I was surprised when reading the results, because the reported analyses do not seem to follow the conceptual framing of the ms. For instance, “sex” is being treated as a confounding factor in several analyses, although even the title of the ms. suggests that sex is a focal variable in the present context. In this vein, the results section seems to be only loosely connected to the described research focus of the introduction. The structure of this section needs to be considerably improved to allow a meaningful evaluation of this study’s merit.

-        Especially when considering the suboptimal power of the present study, it would be useful to focus on the interpretation of effect sizes (e.g., Cohen d; please add respective values) instead of nominal null hypothesis significance testing.

-        The first para of the Discussion should be removed. I acknowledge the authors intent to summarize the most important results, but in the presented manner, this para does not succeed in providing the (I presume) desired effect of structuring the results in an easily apprehensible manner – it is a mere reiteration of some selected (but only loosely related) results.

-        The Discussion could be shortened and its structure should be improved. More emphasis needs to be placed on the meaning of the presently observed results as opposed to lengthy descriptions of previous studies and speculations about possible implications that are largely disconnected from the present work.

-        I was surprised about the concluding para of the ms. which seemingly came out of nowhere and suggested that there is a sex-specific protective effect of giftedness against peer problems and social behaviors which I had not yet seen discussed explicitly in the entire Discussion section.

-        The ms. needs to be carefully revised to avoid awkward phrasing and misleading wording (e.g., lines 139-140 read as if only children but no caregivers were asked for consent).

-        The authors use the terms sex and gender interchangeably, which should be avoided. According to APA and AMA publication manuals, the term “sex” should be used when referring to biological characteristics, whilst “gender” is used to describe sex role orientation.

-        In the spirit of transparency and open science practices, I would like to encourage the authors to provide both data and analysis codes as supplementary materials or in an online repository.

Comments on the Quality of English Language

The ms. should be carefully rechecked to weed out awkward wording and phrasing..

Author Response

Thank you for your comprehensive and insightful review of this paper; your critical perspective and constructive suggestions have contributed significantly to the further development of our work.

1- The authors describe, that stage three of the giftedness assessment is based on “a valid, reliable, and standardized intelligence test”, namely the Anadolu-Sak intelligence scale. Most readers of this paper will never have heard about this scale (at the very least, I haven’t) and providing a non-dated reference (line 99) to a test manual/book chapter (???) is not informative either. This is a minor point, because in case of test unreliability or non-validity of the used test, any real existing effect would become harder to detect due to the increase of the signal-to-noise ratio. However, the authors should fill the reader in at least about the defining characteristics of this test and provide a proper citation (please also provide information about the used test norms including the standardization year).

AU: Thank you for your suggestion, Anadolu Sak Intelligence Scale were added to the measures section. It is presented in the following.

“2.2.3. Anadolu Sak Intelligence Scale

The Anadolu-Sak Intelligence Scale (ASIS) is an individually implemented intelligence test battery developed for Turkish-speaking children aged four to 12 and normed in Turkey in 2016 29,37. ASIS is theoretically based on the Cattell-Horn-Carroll model of intelligence and cognitive abilities 38. It consists of 256 items, seven subtests, and three factors. Several researchers have investigated the reliability and validity of the ASIS 29,37,39. The validity of the ASIS was studied and confirmed using exploratory and confirmatory factor analyses. Sak et al. determined correlations between ASIS scores and the UNIT and the RIAS intelligence test scores ranging from .50 to .82 29. Studies investigating the reliability of ASIS demonstrate that the internal consistency of the subtests and the component scores range from .81 to .94 in the standardization sample 40. Inter-coder reliability has been reported at .96 and above 29. Test-retest reliability for the subtests ranges from .66 to .85, and between .91 and .95 for the factors 39.”

2- Please provide Cronbach alpha from your test administrations of both SDQ and TOM questionnaires/tests.

AU: Thank you for your contribution. The Cronbach's alpha values of peer problems (0.72), prosocial behavior (0.80) scales and RMET-C (0.73), FPRT-C (0.77) tests were added to the measures section.

3- Move participant descriptives from the results to the participants section.

AU: “The study population consisted of 45 gifted children (22 boys, 23 girls) and 45 non-gifted children (18 boys, 27 girls). Mean ages were (11.6±0.6 years) in the gifted group and (11.5±0.5 years) in the non-gifted group.” this sentence was removed from the results section and added to the participants section.

4-        You should provide the readers with the information in which way groups differed from each other when describing results. It is not enough to indicate that there were differences if the effect direction (e.g., larger values of gifted vs. non-gifted individuals) is not provided.

AU: Thank you for your suggestion. The result section has edited according to your suggestion.

“No significant sex or age differences were observed between the gifted and non-gifted groups. Peer problem scores were statistically significantly higher in the non-gifted group than in the gifted group, but no significant difference was found in prosocial behavior scores. Significantly fewer peer problems and more prosocial behaviors were determined in boys from the gifted group than in those from the non-gifted group, but no significant difference was found between the girls from two groups. ToM tests were statistically significantly higher in boys from the gifted group than in those from the non-gifted group, only FPRT-C scores were higher in girls from the gifted group. ToM test results were statistically significantly higher in the gifted group than in the non-gifted group (Table 1).

Girls from the non-gifted group exhibited significantly lower peer problem scores and higher prosocial behavior, RMET-C, and FPRT-C scores than boys, although no significant differences were determined between boys and girls from the gifted group (Table 2).

The group effect (F1,86=8.237, p=0.005) had a significant effect on peer problems, while sex (F1,86=3.301, p=0.073) was not significant. The interaction of group and sex (F1,86=3.991, p=0.049) also exerted a significant effect on peer problems. The gifted group registered lower peer problem scores. In the non-gifted group, peer problems were higher in the boys. Sex (F1,86=4.300, p=0.041) had a significant effect on prosocial behavior scores, while the group effect (F1,86=0.613, p=0.436) was not significant. Group and sex interaction (F1,86=5.837, p=0.018) resulted in significant differences in prosocial behavior scores. Girls registered higher prosocial behavior scores. In the non-gifted group, the prosocial behavior score was lower among the boys. The group effect (F1,86=8.374, p=0.005) and sex (F1,86=5.148p=0.026) significant affected RMET-C scores. Hoever, the interaction of group and sex (F1,86=1.563, p=0.215) was not significant on RMET-C scores. The group effect (F1,86=26.832, p<0.001) and sex (F1,86=10.127, p= 002) exerted significant effects on FPRT-C scores, although the interaction of group and sex (F1,86=1.923, p=0.169) was not significant on FPRT-C scores. ToM test scores were higher in the gifted group and in girls (Table 3).”

5- I was surprised when reading the results, because the reported analyses do not seem to follow the conceptual framing of the ms. For instance, “sex” is being treated as a confounding factor in several analyses, although even the title of the ms. suggests that sex is a focal variable in the present context. In this vein, the results section seems to be only loosely connected to the described research focus of the introduction. The structure of this section needs to be considerably improved to allow a meaningful evaluation of this study’s merit.

AU:Thank you for your valuable suggestion. In our study, we omitted ANCOVA, which measures the confounding effect of gender, and instead used two-way ANOVA to evaluate the main and interaction effects of the independent variables. The results of the two-way ANOVA are presented below.

“The group effect (F1,86=8.237, p=0.005) had a significant effect on peer problems, while sex (F1,86=3.301, p=0.073) was not significant. The interaction of group and sex (F1,86=3.991, p=0.049) also exerted a significant effect on peer problems. The gifted group registered lower peer problem scores. In the non-gifted group, peer problems were higher in the boys. Sex (F1,86=4.300, p=0.041) had a significant effect on prosocial behavior scores, while the group effect (F1,86=0.613, p=0.436) was not significant. Group and sex interaction (F1,86=5.837, p=0.018) resulted in significant differences in prosocial behavior scores. Girls registered higher prosocial behavior scores. In the non-gifted group, the prosocial behavior score was lower among the boys. The group effect (F1,86=8.374, p=0.005) and sex (F1,86=5.148p=0.026) significant affected RMET-C scores. Hoever, the interaction of group and sex (F1,86=1.563, p=0.215) was not significant on RMET-C scores. The group effect (F1,86=26.832, p<0.001) and sex (F1,86=10.127, p= 002) exerted significant effects on FPRT-C scores, although the interaction of group and sex (F1,86=1.923, p=0.169) was not significant on FPRT-C scores. ToM test scores were higher in the gifted group and in girls (Table 3).”

6- Especially when considering the suboptimal power of the present study, it would be useful to focus on the interpretation of effect sizes (e.g., Cohen d; please add respective values) instead of nominal null hypothesis significance testing.

Thank you for your suggestion. Effect size (Cohen's d and Cramer's V) was added to the results section. The two-way ANOVA partial eta squared value was also presented.

7- The first para of the Discussion should be removed. I acknowledge the authors intent to summarize the most important results, but in the presented manner, this para does not succeed in providing the (I presume) desired effect of structuring the results in an easily apprehensible manner – it is a mere reiteration of some selected (but only loosely related) results.

Thank you for your suggestion. The first paragraph of the discussion section was simplified and reorganized. It is presented in the following.

“ToM test results in this study were higher among the gifted children and girls. Peer problems were lower among the gifted children but higher in boys from the non-gifted group, while no sex difference was observed among the gifted children. Prosocial behavior was higher in girls. While no sex differences were observed in the gifted children, prosocial behavior was lower in boys from the non-gifted group. No relationship was determined between ToM tests and peer problems or prosocial behavior in the gifted children, but such an association was observed in the non-gifted group.”

8-  The Discussion could be shortened and its structure should be improved. More emphasis needs to be placed on the meaning of the presently observed results as opposed to lengthy descriptions of previous studies and speculations about possible implications that are largely disconnected from the present work.

AU: Thank you for your valuable suggestion. The following parts of the discussion section have removed. The structure of the discussion section is reorganised and improved.

“Children with enhanced ToM skills better understand their own and others' emotions, which adds to their social functioning 43 High IQ contributes to finding valuable strategies for coping with difficulties 44,45.”

“Gifted girls scored higher on the FPRT-C test than non-gifted girls, while there was no significant difference between the groups’ RMET-C test results. The gifted boys scored higher on both ToM tests.”

“Francis et al. found that gifted children exhibited better socio-emotional adjustment and fewer behavioral problems than their peers 48. Chung et al. reported that gifted adolescents exhibited more cooperative behaviors toward their peers 49. Gifted children have also been described as popular among their peers 50. A study of adolescents aged between 12 and 16 reported social-emotional problems 51. Eren et al. compared gifted children and adolescents across a wide age range (9-18 years) with a non-gifted group and found no difference in the parents’ prosocial behavior and peer relationship scores 52.”

"No difference in peer probles and prosocial behavior was observed among girls in the present research, indicating their general superiority in these areas. However, the fact that gifted boys had fewer peer problems and exhibited better prosocial behavior suggests that giftedness is a protective factor among boys.”

9-  I was surprised about the concluding para of the ms. which seemingly came out of nowhere and suggested that there is a sex-specific protective effect of giftedness against peer problems and social behaviors which I had not yet seen discussed explicitly in the entire Discussion section.

AU: The part suggesting that giftedness has a sex-specific protective effect against peer problems and social behaviours was removed from the conclusion paragraph. The statements in the discussion section and the conclusion section were reorganised.

10- The ms. needs to be carefully revised to avoid awkward phrasing and misleading wording (e.g., lines 139-140 read as if only children but no caregivers were asked for consent).

AU: The text was revised again. The relevant lines have been edited as follows.

Written informed consent forms were obtained from children and parents.

11- The authors use the terms sex and gender interchangeably, which should be avoided. According to APA and AMA publication manuals, the term “sex” should be used when referring to biological characteristics, whilst “gender” is used to describe sex role orientation.

AU: Thank you for your suggestion. Since our research is based on the biological sex of children (male or female), it is more accurate to use the term 'sex'.

12-   In the spirit of transparency and open science practices, I would like to encourage the authors to provide both data and analysis codes as supplementary materials or in an online repository.

AU: The Science and Arts Center (SAC) pull-out enrichment collaboration restricts us from sharing data as open access. The data that support the findings of this study are available on request from the corresponding author.

13- The ms. should be carefully rechecked to weed out awkward wording and phrasing.

AU: The article was carefully checked again.

Reviewer 2 Report

Comments and Suggestions for Authors

This study presents analysis of the effects of giftedness and sex differences on children’s theory of mind skills and social behavior, comparing the high-level ToM skills of gifted children with those of their peers with typical development.

Comments

Rethink the use of gender, instead of sex (birth sex), since gender refers to the socially constructed characteristics of women and men.

Clarify what is the participants age range, abstract and participants section are different.

Page 2, lines 66-72, authors should discuss methodological differences among cited studies.

Page 3, lines 101-102, authors should whether non-gifted participants are above average of most children, since they were nominated by their classroom teacher to take a screening exam. There are implications for generalization if these is true.

Months and years of data collection should be stated.

For the Statistical Analysis, in the title and in the last paragraph of Introduction, authors stated investigating the effects of giftedness and sex differences. I failed to understand why not use sex as independent variable instead of as confounder. Authors should be able to show giftedness and sex main effects and two-way interaction by using ANOVA two-way for each dependent variable (peer, prosocial, RMET-C and FPRT-C). Also, they will be able to rewrite the results, constructing one paragraph for each dependent variable, instead tell us, for instance, about peer problems mixed with the rest.

In the Results section, there are many data from tables in the paragraphs. I believe that there is no need for write in the text results that are shown in Tables. If analyses are improved, results can be clearer.

Table 1 formatting makes it very hard to understand; what is the meaning of NA; and Chi-square symbol is a Greek letter. Also, in most tables, authors are presenting “mean +SD” or “mean + 95%CI”?

In Table 4, I failed to understand the point of analyzing correlations for the total sample, given all the differences reported before this table. Additionally, I missed correlations separated by sex.

Only 17% of the cited references are within the last 5 years.

Comments on the Quality of English Language

The use of some words is indicated in the PDF file.

Author Response

Thank you for your comprehensive and insightful review of this paper; your critical perspective and constructive suggestions have contributed significantly to the further development of our work.

1- Rethink the use of gender, instead of sex (birth sex), since gender refers to the socially constructed characteristics of women and men.

AU: Thank you for your suggestion. Since our research is based on the biological sex of children (male or female), it is more accurate to use the term 'sex'.

2- Clarify what is the participants age range, abstract and participants section are different.

AU: Thank you for your suggestion. Age corrected in abstract section

3-Page 2, lines 66-72, authors should discuss methodological differences among cited studies.

AU: Thank you for your suggestion. The following section is added to the relevant part of the article.

“The fact that in terms of methodology the researchers methodologically used self-generated questions, valid and reliable scales, and parent or self-report scales in those studies may account for the differences in the findings.”

4- Page 3, lines 101-102, authors should whether non-gifted participants are above average of most children, since they were nominated by their classroom teacher to take a screening exam. There are implications for generalization if these is true.

AU: Thank you for your valuable suggestion. We selected the control group from the normal class sample. The majority of the group consists of students who are not candidates for the screening exam. This is because only about 10% of each class is nominated for the screening exams. Considering this low percentage and the randomization of the control group, we think that the control group performs at an average level. The relevant lines were edited as follows.

“The non-gifted population in this study consisted of students who were not nominated by their classroom teachers for tests of SACs or nominated by scoring below 130 on the ASIS test.”

5-Months and years of data collection should be stated.

AU: The following sentence was added to Procedure and Data Collection.

“The data were collected between 9 September and 10 October, 2023.”

6- For the Statistical Analysis, in the title and in the last paragraph of Introduction, authors stated investigating the effects of giftedness and sex differences. I failed to understand why not use sex as independent variable instead of as confounder. Authors should be able to show giftedness and sex main effects and two-way interaction by using ANOVA two-way for each dependent variable (peer, prosocial, RMET-C and FPRT-C). Also, they will be able to rewrite the results, constructing one paragraph for each dependent variable, instead tell us, for instance, about peer problems mixed with the rest.

AU:Thank you for your valuable suggestion. In our study, we omitted ANCOVA, which measures the confounding effect of gender, and instead used two-way ANOVA to evaluate the main and interaction effects of the independent variables. The results of the two-way ANOVA are presented below.

“The group effect (F1,86=8.237, p=0.005) had a significant effect on peer problems, while sex (F1,86=3.301, p=0.073) was not significant. The interaction of group and sex (F1,86=3.991, p=0.049) also exerted a significant effect on peer problems. The gifted group registered lower peer problem scores. In the non-gifted group, peer problems were higher in the boys. Sex (F1,86=4.300, p=0.041) had a significant effect on prosocial behavior scores, while the group effect (F1,86=0.613, p=0.436) was not significant. Group and sex interaction (F1,86=5.837, p=0.018) resulted in significant differences in prosocial behavior scores. Girls registered higher prosocial behavior scores. In the non-gifted group, the prosocial behavior score was lower among the boys. The group effect (F1,86=8.374, p=0.005) and sex (F1,86=5.148p=0.026) significant affected RMET-C scores. Hoever, the interaction of group and sex (F1,86=1.563, p=0.215) was not significant on RMET-C scores. The group effect (F1,86=26.832, p<0.001) and sex (F1,86=10.127, p= 002) exerted significant effects on FPRT-C scores, although the interaction of group and sex (F1,86=1.923, p=0.169) was not significant on FPRT-C scores. ToM test scores were higher in the gifted group and in girls (Table 3).”

7- In the Results section, there are many data from tables in the paragraphs. I believe that there is no need for write in the text results that are shown in Tables. If analyses are improved, results can be clearer.

Thank you for your suggestion. The data in the tables have removed from the paragraphs in the results section.

8-  Table 1 formatting makes it very hard to understand; what is the meaning of NA; and Chi-square symbol is a Greek letter. Also, in most tables, authors are presenting “mean +SD” or “mean + 95%CI”?

AU: Table 1 was probably changed when the article was designed according to the journal's draft. NA removed. Corrected chi-squared sign, presented as "mean + SD" in most tables.

9- In Table 4, I failed to understand the point of analyzing correlations for the total sample, given all the differences reported before this table. Additionally, I missed correlations separated by sex.

AU: Thank you for your suggestion, the total sample correlation has removed from the table.

10-Only 17% of the cited references are within the last 5 years.

AU: During the revision process, we have made an effort to increase this rate by referring to publications in recent years. 

Reviewer 3 Report

Comments and Suggestions for Authors

The manuscript examines the relationship between Theory of Mind (ToM) abilities and social behavior in gifted and non-gifted children and shows that gifted children with high cognitive abilities also have superior social cognition skills.

Overall, the work is noteworthy, and I think the topic of this study may be of interest to journal readers. However, I think that the authors should resolve the following issues in order for the manuscript to be considered worthy of publication.

Title: Since this is a cross-sectional study and the direction of the associations found cannot be determined, it would be better to title the paper "The relationship between giftedness..." instead of “The impacts of giftedness...."

Materials and Methods:

- The study was conducted with 45 gifted and 45 non-gifted children. How was the number of subjects to be recruited for the study calculated? Was a power analysis carried out?

- Intelligence was assessed using the Anadolu-Sak Intelligence Scale (ASIS). As this test is scarcely used in the literature, further information is needed to attest its psychometric validity.

- When the authors cite the RMET, they refer to a version ([25] Baron-Cohen et al., 1997) that was validated for adults and not for children and that was later replaced by a revised version (Baron-Cohen et al., 2001, PMID: 11280420). On which version was the Turkish version cited as reference [26] conducted?

- line 125: "...when an individual says something he OR SHE should not say..."

Results:

- Were the comparisons presented in this section correct for multiple comparisons? If not, why?

- Table 1 is very difficult to read

Discussion:

- In lines 288-289, the authors state that the study included a heterogeneous sample representing gifted children in different ability areas, but not enough information is provided to know what these different ability areas are.

- The main limitation of the study is the lack of an assessment of executive functions. The authors acknowledge that “Social behaviors in gifted children may be related to other cognitive abilities, such as executive functions..." but provide no explanation as to why these abilities were not assessed. This must clearly be mentioned as a limitation of the work, also in view of the fact that the literature is divided regarding the relationship between ToM and executive functions. For example, Di Tella et al. (2020, PMID: 33057089) showed a lack of association between affective/cognitive ToM and executive functions, while Stone and Gerrans (2006, PMID: 18633796) claim the opposite. The authors should refer to this literature and emphasize that further studies in populations of gifted children could help to clarify this aspect.

Author Response

Thank you for your comprehensive and insightful review of this paper; your critical perspective and constructive suggestions have contributed significantly to the further development of our work.

1-Title: Since this is a cross-sectional study and the direction of the associations found cannot be determined, it would be better to title the paper "The relationship between giftedness..." instead of “The impacts of giftedness...."

AU: Thank you for your suggestion. The title changed to the following.

“The Relationship between Giftedness and Sex and Children’s Theory of Mind Skills and Social Behavior”

Materials and Methods:

2- - The study was conducted with 45 gifted and 45 non-gifted children. How was the number of subjects to be recruited for the study calculated? Was a power analysis carried out?

AU: The effect size (Cohen's d) was calculated as 0.64, with 80% power and a 95% confidence interval, analysis showing that 40 children in each group should be included in the study 41. Based on a 10% probability of dropouts in each group, a minimum of 88 participants with 44 patients in each group was required.

3- Intelligence was assessed using the Anadolu-Sak Intelligence Scale (ASIS). As this test is scarcely used in the literature, further information is needed to attest its psychometric validity.

AU: Thank you for your suggestion, Anadolu Sak Intelligence Scale were added to the measures section. It is presented in the following.

 “The Anadolu-Sak Intelligence Scale (ASIS) is an individually implemented intelligence test battery developed for Turkish-speaking children aged four to 12 and normed in Turkey in 2016 29,37. ASIS is theoretically based on the Cattell-Horn-Carroll model of intelligence and cognitive abilities 38. It consists of 256 items, seven subtests, and three factors. Several researchers have investigated the reliability and validity of the ASIS 29,37,39. The validity of the ASIS was studied and confirmed using exploratory and confirmatory factor analyses. Sak et al. determined correlations between ASIS scores and the UNIT and the RIAS intelligence test scores ranging from .50 to .82 29. Studies investigating the reliability of ASIS demonstrate that the internal consistency of the subtests and the component scores range from .81 to .94 in the standardization sample 40. Inter-coder reliability has been reported at .96 and above 29. Test-retest reliability for the subtests ranges from .66 to .85, and between .91 and .95 for the factors 39.”

4- When the authors cite the RMET, they refer to a version ([25] Baron-Cohen et al., 1997) that was validated for adults and not for children and that was later replaced by a revised version (Baron-Cohen et al., 2001, PMID: 11280420). On which version was the Turkish version cited as reference [26] conducted?

AU: The RMET Turkish version study was conducted in both adults and children. The child study group consisted of 202 normally developing children and 33 children with autism aged 6-16 years. According to the results, the child form had adequate internal consistency reliability (Cronbach's Alpha value for the child form was 0.72).

5- line 125: "...when an individual says something he OR SHE should not say..."

AU: Thank you for the correction.  The sentence was corrected.

Results

6-  Were the comparisons presented in this section correct for multiple comparisons? If not, why?

AU: Appropriate for multiple comparisons, research outputs were re-evaluated using a two-factor ANOVA.

7-  Table 1 is very difficult to read

AU: Table 1 corrected.

Discussion

8- In lines 288-289, the authors state that the study included a heterogeneous sample representing gifted children in different ability areas, but not enough information is provided to know what these different ability areas are.

AU: The relevant paragraph is revised as follows.

“The strong points of this study are that the age range was narrow, sex differences were evaluated, the groups were composed of individuals with similar sociocultural characteristics, and the study involved a heterogeneous sample representing gifted children in various areas of ability, including science, mathematics, technology, and arts.”

9-  - The main limitation of the study is the lack of an assessment of executive functions. The authors acknowledge that “Social behaviors in gifted children may be related to other cognitive abilities, such as executive functions..." but provide no explanation as to why these abilities were not assessed. This must clearly be mentioned as a limitation of the work, also in view of the fact that the literature is divided regarding the relationship between ToM and executive functions. For example, Di Tella et al. (2020, PMID: 33057089) showed a lack of association between affective/cognitive ToM and executive functions, while Stone and Gerrans (2006, PMID: 18633796) claim the opposite. The authors should refer to this literature and emphasize that further studies in populations of gifted children could help to clarify this aspect.

AU:Thank you for your suggestion. Your suggestions were added to the article as follows.

“Two different areas of cognition have been defined as hot and cold cognition from the perspective of understanding cognitive processes 57. In the present study, ToM tests mainly assessed hot cognition, and cold cognition areas, such as executive functions (e.g., working memory) were not examined. The literature emphasizes the relationship between social behavior and other cognitive abilities, such as executive functions 54. At the same time, Di Tella et al. determined no relationship between emotional/cognitive ToM and executive functions, while Stone and Gerrans espoused the opposite view 58,59. Further studies evaluating ToM skills and executive functions in gifted children may help clarify this issue.”

Reviewer 4 Report

Comments and Suggestions for Authors

The manuscript is about The Impacts of Giftedness and Sex on Children’s Theory of 2 Mind Skills and Social Behavior

The study found that peer problems were different in the gifted and non-gifted groups, but prosocial behaviour was similar. The RMET-C and FPRT-C ToM tests revealed significant differences between the gifted and non-gifted groups.

This study sheds light on the fact that gifted children with high cognitive abilities also have superior social cognition skills. Advanced ToM skills in gifted children may be important to support their social and cognitive development.

The introduction and the discussion need to be strengthened, so I indicate below some elements that the authors should improve.

It is important that the authors develop the concept of prosocial behaviour and deepen the educational interventions on prosocial behaviour and empathy in gifted students.

Authors should explicitly include, in the introduction, the question to be answered in the manuscript.

The method is well described.

In the discussion, the suggestions I have included to strengthen the introduction should also be taken into account.

The authors should differentiate the conclusions from the discussion.

Proposals for the future should be pointed out.

Conclusions are convincing with the evidence and arguments presented.

Citations and references are appropriate and up to date, but authors should review the journal's rules, so that they are as requested in Children.

Tables are very clear, but should be checked to ensure that they are in the format requested by the journal in which they are to be published.

The data are reproducible.

The authors contribute knowledge to science with this manuscript, being relevant and of interest.

Author Response

Thank you for your comprehensive and insightful review of this paper; your critical perspective and constructive suggestions have contributed significantly to the further development of our work.

Reviewer 4

1- It is important that the authors develop the concept of prosocial behaviour and deepen the educational interventions on prosocial behaviour and empathy in gifted students.

AU: Thank you for your suggestion. The following section was added to the introduction of the article.

“Prosocial behavior, a social skill which includes actions intended to benefit others, has become a topic of interest in various fields. Research has shown that prosocial behavior is influenced by various factors, such as empathy, genetic predispositions, and social contexts 5–7. Findings also highlight the positive effects of prosocial behavior on children's academic achievement, cognitive development, and social relationships 8–10. Gifted children have been shown to exhibit high levels of empathy and sensitivity, thus contributing to their prosocial behaviors 11. Promoting empathy and prosocial behaviors in gifted children is regarded as essential for their social and emotional development 8.”

2- Authors should explicitly include, in the introduction, the question to be answered in the manuscript.

AU: Thank you for your suggestion. The relevant section was edited as follows.

“The main purpose of this study was to investigate the relationship between intelligence and sex in children using ToM and social behavior measures. The specific aim was to determine differences in ToM and social behavior by comparing gifted and non-gifted children. We also set out to evaluate whether there are sex differences in ToM and social behavior between gifted and non-gifted children and to determine the relationships between ToM and social behavior in both groups. We hypothesized that gifted children would exhibit better ToM skills and social behavior than non-gifted children and that girls would exhibit better ToM and social behavior than boys.”

3- The method is well described.

4- In the discussion, the suggestions I have included to strengthen the introduction should also be taken into account.

AU: Thank you for your suggestion. The following section was added to the introduction of the article. (The suggestions of other reviewers have strengthened the introduction and discussion section.)

“Programs aimed at developing empathy skills in gifted children have been shown to yield positive results 49. Since prosocial behaviors play a crucial role in social adjustment to and academic success in school 50, the education of gifted students should be reinforced with high prosocial behaviors exhibited by gifted individuals.”

5- The authors should differentiate the conclusions from the discussion.

AU: Thank you for your suggestion. The first sentence of the discussion was reorganized and important findings were emphasized. It was also removed from other sentences in the discussion.

6- Proposals for the future should be pointed out.

AU: Thank you for your suggestion. The following sentence was added to the discussion section of the article.

“Further studies evaluating ToM skills and executive functions in gifted children may help clarify this issue.”

7- Citations and references are appropriate and up to date, but authors should review the journal's rules, so that they are as requested in Children.

AU: Citations and references were reviewed.

8- Tables are very clear, but should be checked to ensure that they are in the format requested by the journal in which they are to be published.

The tables were reviewed again.

9-The data are reproducible.

AU: Thank you for your comments.

Round 2

Reviewer 1 Report

Comments and Suggestions for Authors

Without a doubt, the authors made an effort to improve the ms. However, I would like them to - once again - reconsider their framing of their findings in terms of effect sizes instead of mere significance. Although they added effect size estimates to their tables, the results are virtually exclusively framed on Null-hypothesis significance testing. It would make more sense to interpret effect meaningfulness instead of mere significance.

I do not understand why anonymized primary data cannot be provided for the present ms.

Comments on the Quality of English Language

minor editing required.

Author Response

1. Without a doubt, the authors made an effort to improve the ms. However, I would like them to - once again - reconsider their framing of their findings in terms of effect sizes instead of mere significance. Although they added effect size estimates to their tables, the results are virtually exclusively framed on Null-hypothesis significance testing. It would make more sense to interpret effect meaningfulness instead of mere significance.

AU: Thank you for taking the time to review our work and for your valuable feedback. In line with your suggestion, the article has been framed in terms of effect sizes and revised again. Your suggestions are included in the article as follows.

"Results

Peer problem scores were statistically significantly higher in the non-gifted group than in the gifted group at medium effect size, but no significant difference was found in prosocial behavior scores. Boys in the gifted group had significantly fewer peer problems (large effect size) and more prosocial behaviors (medium effect size) than boys in the non-gifted group, while no significant difference was found between the girls in the two groups. ToM tests were statistically significantly higher for boys in the gifted group than in the non-gifted group at large effect size; only the FPRT-C scores were higher for girls in the gifted group. ToM test results were statistically significantly higher in the gifted group than in the non-gifted group at medium and large effect sizes (Table 1).

            The group effect (F1,86=8.237, p=0.005) had a medium significant effect on peer problems, while sex (F1,86=3.301, p=0.073) was not significant. The interaction of group and sex (F1,86=3.991, p=0.049) also exerted a small to medium significant effect on peer problems. The gifted group registered lower peer problem scores. In the non-gifted group, peer problems were higher in the boys. Sex (F1,86=4.300, p=0.041) had a small to medium significant effect on prosocial behavior scores, while the group effect (F1,86=0.613, p=0.436) was not significant. Group and sex interaction (F1,86=5.837, p=0.018) resulted in medium significant differences in prosocial behavior scores. Girls registered higher prosocial behavior scores. In the non-gifted group, the prosocial behavior score was lower among the boys. The group effect (F1,86=8.374, p=0.005) and sex (F1,86=5.148p=0.026) medium significant affected RMET-C scores. However, the interaction of group and sex (F1,86=1.563, p=0.215) was not significant on RMET-C scores. The group effect (F1,86=26.832, p<0.001) and sex (F1,86=10.127, p= 002) exerted large and medium significant effects on FPRT-C scores, respectively, although the interaction of group and sex (F1,86=1.923, p=0.169) was not significant on FPRT-C scores. ToM test scores were higher in the gifted group and in girls (Table 3).

Discussion

The higher medium and large effect sizes of the ToM test results in gifted children indicate that this difference is practically significant.

This study's finding that ToM test results were higher in the large effect size for girls reinforces the previous literature.

The present study's finding of higher prosocial behaviors in gifted boys with small to medium effect sizes should be confirmed in a larger sample or with further studies."

2. I do not understand why anonymized primary data cannot be provided for the present ms.

AU: We understand your concern about sharing anonymized primary data and fully recognize the importance of open science practices. Unfortunately, this is the Science and Arts Center recommendation. Thank you for your understanding.

Reviewer 3 Report

Comments and Suggestions for Authors

The authors have answered all the points I raised in the first version of the article. 

Author Response

Thank you for taking the time to review our work and for your valuable feedback. 

Reviewer 4 Report

Comments and Suggestions for Authors

The authors have enriched the manuscript and have addressed many of the suggestions indicated by the reviewers. But the manuscript needs to be strengthened with some systematic review about educational interventions in prosocial behavior and empathy in students with high abilities. 

Author Response

The authors have enriched the manuscript and have addressed many of the suggestions indicated by the reviewers. But the manuscript needs to be strengthened with some systematic review about educational interventions in prosocial behavior and empathy in students with high abilities. 

AU: Thank you for your valuable suggestions. We could not find a systematic review in the literature that directly includes educational interventions for prosocial behavior and empathy in high ability students. However, we fulfilled your suggestion with systematic review / meta-analysis / research studies on educational interventions for prosocial behavior and empathy in the literature. Your suggestion was added to the article as follows.

“Intervention programs aimed at promoting prosocial behaviors in children were found to show significant effectiveness for these behaviors 12,13. A meta-analysis study reported that programs targeting prosocial behavior and empathy-related skills were more effective on social and academic outcomes when implemented at earlier ages and included more empathy skills 14. The empathy training program was found to have lasting effects on gifted adolescents with low empathy scores 15. In a recent systematic review, experiential learning programs among learning programs were determined to be more effective in developing empathy 16. A systematic review by Cheang et al. suggested mindfulness-based interventions increase empathy in children and adolescents 17. Implementing these programs for psychosocial competence in gifted children may be helpful.”